# Drug-dependent growth curve reshaping reveals mechanisms of antifungal resistance in *Saccharomyces cerevisiae*

Lesia Guinn [1,2], Evan Lo [1] & Gábor Balázsi [1,2✉]

Microbial drug resistance is an emerging global challenge. Current drug resistance assays tend to be simplistic, ignoring complexities of resistance manifestations and mechanisms, such as multicellularity. Here, we characterize multicellular and molecular sources of drug resistance upon deleting the *AMN1* gene responsible for clumping multicellularity in a budding yeast strain, causing it to become unicellular. Computational analysis of growth curve changes upon drug treatment indicates that the unicellular strain is more sensitive to four common antifungals. Quantitative models uncover entwined multicellular and molecular processes underlying these differences in sensitivity and suggest *AMN1* as an antifungal target in clumping pathogenic yeasts. Similar experimental and mathematical modeling pipelines could reveal multicellular and molecular drug resistance mechanisms, leading to more effective treatments against various microbial infections and possibly even cancers.

[1] The Louis and Beatrice Laufer Center for Physical and Quantitative Biology, Stony Brook University, Stony Brook, NY 11794, USA. [2] Department of Biomedical Engineering, Stony Brook University, Stony Brook, NY 11794, USA. ✉email: gabor.balazsi@stonybrook.edu

Microbial drug resistance is a major global challenge[1]. Despite our vast knowledge of its molecular mechanisms, and rapid discovery of antibiotics[2] we are still far from predicting or effectively stopping drug resistance, possibly because its underlying processes may transcend purely molecular phenomena. For example, drug resistance can involve various multicellular mechanisms that surpass molecular interactions in many different settings – from bacterial biofilms colonizing medical implant devices[3–6] to circulating tumor cell clusters[7,8]. Nonetheless, multicellularity has molecular underpinnings, making the phenotypic effects of multicellularity and its molecular roots difficult to disentangle.

Yeast pathogens can present a variety of multicellular phenotypes (biofilms, flocs, chains, and clumps)[9] that withstand generic environmental stressors[10,11] and antifungals[12–15]. As opposed to multidrug transporter-mediated, purely molecular drug elimination[16,17], these multicellular structures can spatially reduce the penetration of drugs or other stressors, facilitating short-term survival, and subsequent long-term evolutionary adaptation by various resistance mechanisms[18,19]. Yeast biofilms, mats[20,21] attached to surfaces or flocs[22–25] formed in suspension via cell wall-mediated, non-clonal cell aggregation, can provide resistance to various stressors[25]. Unicellular yeast in suspension can also evolve into[26] or back from[27,28] clumping, a non-flocculating form of multicellularity that stems from failed daughter-mother cell separation. Work by others[29–33] and us[27] indicates that clumping is orchestrated by the mitotic exit network (MEN), a transcriptional regulatory program driven by the mitotic inducer *ACE2* and its downstream target mitotic antagonist gene *AMN1*. Like flocculation, clumping seems to provide environmental stress resistance[27,28], yet such effects could also stem from pleiotropic effects of *AMN1* unrelated to clumping. Thus, while yeast clumping is emerging as a model for testing, quantifying, and interpreting resistance to drugs, immunity or environmental stressors in multicellular fungi[27,28], bacteria[34,35] or even cancer cells[8,36], the underlying mechanisms need further exploration.

Detailed, quantitative investigation of time-dependent drug effects on microbes is increasingly important[17,37–40], yet remains insufficient in widely used, traditional experimental approaches. For example, series of photographs and colony counts are common in testing fungal drug sensitivity on solid media[37]. In liquid media, common drug response measures (MIC, minimal inhibitory concentration, and EC, effective concentration) are single numbers[41,42] that ignore other potentially informative parameters, such as the growth inhibition time, the adaptation duration in case of regrowth, and the exponential growth or death rate. Parametrized growth/death curves in stressful conditions[43–46] should be suitable to reveal time-dependent drug resistance characteristics and mechanisms[43], but quantitative analysis and modeling of growth curves, or understanding their implications about multicellularity remain open problems.

Here we establish the genetic basis for clumping multicellularity in TBR1 budding yeast (*S. cerevisiae* Σ1278b) by its conversion to unicellularity upon deleting the gene *AMN1*. We develop quantitative analyses and mathematical modeling to compare how four different antifungals reshape the growth curves of clumping TBR1 and its unicellular *AMN1*-deleted derivative TBR1Δa strain, as well as wild-type and *AMN1*-deleted unicellular S288c lab strains. These analyses uncover that *AMN1* deletion sensitizes TBR1 cells to all antifungals, in drug-specific ways, not just by abrogating clumping, but also by other pleiotropic effects, which remain to be unraveled. The interdisciplinary methods we develop and conclusions we draw should provide a quantitative framework for understanding drug resistance mechanisms in various uni- and multicellular microbes and may guide clinical approaches towards designing improved drugs and therapies.

## Results

**Deleting *AMN1* from clump-forming yeast abrogates multicellularity and accelerates growth.** Clumping in yeast stems from cells unable to separate in mitosis, forming isogenic clusters. Considering the genetic bases of this multicellular phenotype in other strains and settings[27,29,32], we hypothesized that deleting the *AMN1* gene should convert the clumpy haploid yeast TBR1 (*S. cerevisiae* Σ1278b strain 10560-23C; MATα, ura3-52, his3::hisG, leu2::hisG) strain (Fig. 1a) to unicellular in liquid culture. To test this hypothesis and engineer a robustly unicellular strain with minimal genetic difference from TBR1, we designed a homologous recombination-based knock-out cassette with upstream and downstream *AMN1*-complementary sequences (homology arms) flanking the kanamycin resistance gene *KanMX6* (Fig. 1c, Supplementary Figs. 1 and 2a). After confirming that the linearized vector contained no replication modules, we integrated this cassette using standard procedures[47] (Methods). We confirmed cassette integration and the lack of intact *AMN1* by local genomic DNA sequencing (Supplementary Table 1, Supplementary Fig. 2b), thus obtaining the TBR1Δa strain.

While we have previously shown that TBR1 evolves towards unicellularity by *AMN1* mutations[27], whether the *AMN1* deletion alone can abrogate clumping in the TBR1 ancestral background has not been tested. To investigate this, we performed quantitative clump size analysis based on custom microscopy image segmentation (Supplementary Fig. 3), obtaining clump size distributions for three strains (Fig. 1d, e, Methods): TBR1Δa, TBR1 and its previously evolved unicellular derivative TBR1 EvoTop[27] (Fig. 1b). The variance and mean indicated narrower and left-shifted clump size distributions for TBR1 EvoTop and TBR1Δa compared to TBR1. Utilizing image segmentation protocols optimized to detect either clumps or single cells led to similar object diameter distributions for TBR1Δa. The average cell and clump sizes of the parental TBR1 and TBR1Δa held up against the clumping positive control KV38[25] and the unicellular negative control YPH500[47] strains. In the unicellular laboratory strain BY4742, *AMN1* deletion did not alter cell and clump size (Fig. 1d). Importantly, the variance and mean of the TBR1Δa clump size distribution were the lowest among all strains tested, strongly demonstrating unicellularity (Fig. 1d, e).

Considering that multicellularity can be disadvantageous in normal settings, with nutrients but without stress[27,48,49], we next asked whether this holds true for TBR1 and TBR1Δa strains that only differ in the lack of *AMN1*. To characterize the growth kinetics of the two strains without stress, we recorded their optical density ($OD_{600}$) growth curves in the common growth medium YPD (yeast extract, peptone, dextrose) and minimal medium SC (synthetic complete) (Methods) with various glucose contents (0.5, 1, and 2%) (Fig. 1f, g, Supplementary Fig. 4). Indeed, TBR1Δa grew slightly faster in all these media according to the known growth benefits of unicellularity[27], although these effects could stem from *AMN1* interactions unrelated to unicellularity. Mathematical models of sugar utilization indicated sugar-limited growth and fit the cell count estimate data best with an Alee effect[50] in glucose (Supplementary Notes 1 and 2, Supplementary Tables 2 and 3, Supplementary Figs. 4–6) for both strains. With these assumptions, the models captured experimentally observed growth curves while revealing quantitative details of sugar conversion into biomass[51] for the two strains (Supplementary Note 2, Supplementary Table 4).

Overall, we found that *AMN1* deletion is sufficient to cause transition from clumping to unicellular phenotype in TBR1 yeast,

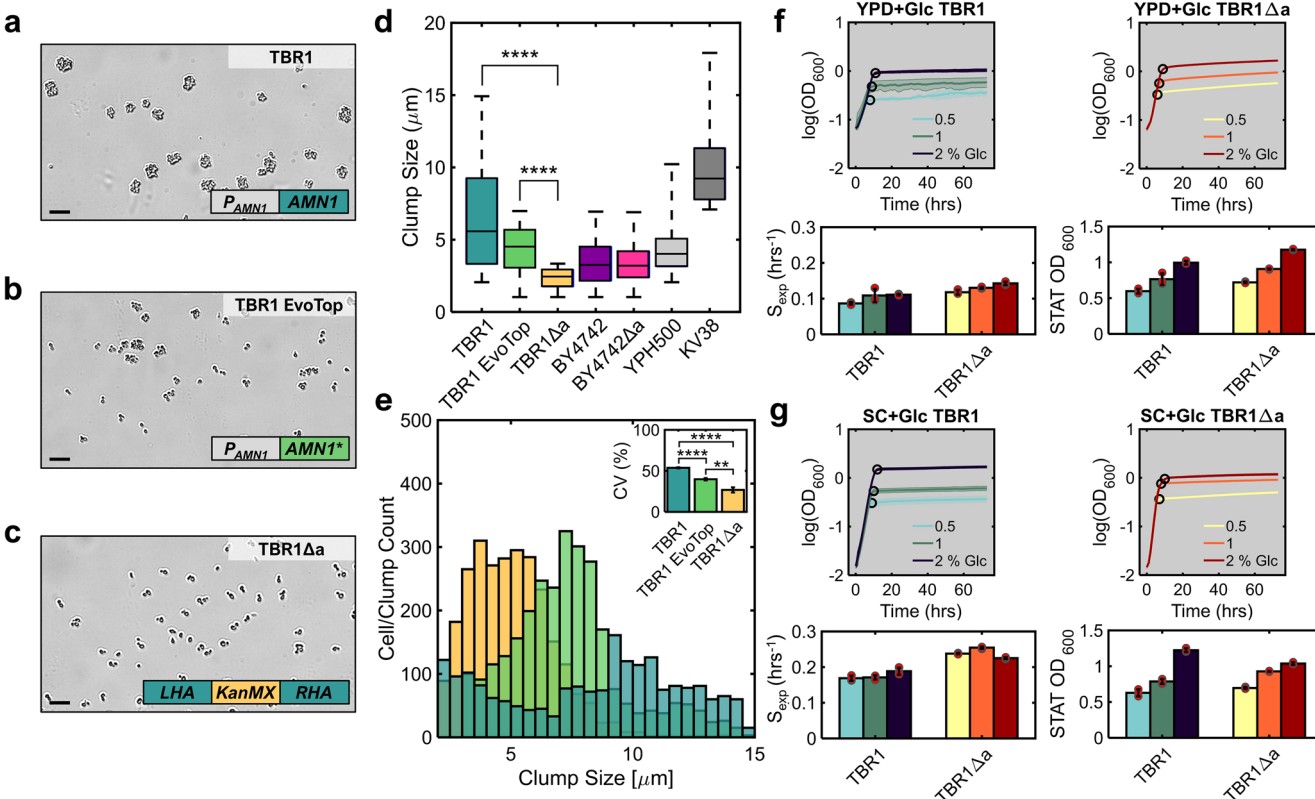

**Fig. 1 *AMN1* knockout causes conversion to unicellularity in S. cerevisiae TBR1 (Σ1278b). a** Brightfield 10x microscope image of clump-forming parental TBR1 strain. The schematic shows the intact native *AMN1* gene expressed from its own promoter. **b** Brightfield 10x microscope image of evolved TBR1 EvoTop cells. The star in the schematic denotes *AMN1* coding sequence mutations that caused partial transition to unicellularity. **c** Brightfield 10x microscope image of *AMN1*-deleted TBR1Δa. The schematic shows the *AMN1* knock-out cassette consisting of left (LHA) and right (RHA) *AMN1* homology arms flanking the kanamycin resistance cassette *KanMX* inserted between the *TEF* promoter and *TEF* terminator. **d** Cell/clump size distributions of TBR1, TBR1 EvoTop, TBR1Δa, BY4742, BY4742Δa, YPH500, and KV38 representative monoclonal populations shown as box plots for $n = 776, 1161, 1591, 1078, 1057, 4128$, and 47 objects, respectively. For statistical analysis, see Supplementary Data 1. **e** Clump/cell size histograms in the TBR1, TBR1 EvoTop, and TBR1Δa strains. Inset in the corner: clump/cell size coefficient of variation (CV, %), defined as the standard deviation, SD, normalized by the mean (calculated from three independent clonal populations). **f, g** Absorbance-based growth curves (mean $OD_{600}$ values) plotted on a semilogarithmic scale with confidence intervals calculated from three replicates of TBR1 (blue) and TBR1Δa (yellow) strains in YPD (**f**) and SC (**g**) media with 0.5, 1, and 2% glucose as carbon source. Black circles indicate the breakpoints defined by piecewise linear fitting. The bar graphs below the growth curves show the corresponding exponential growth rates ($S_{exp}$) and carrying capacities (STAT $OD_{600}$) represented as means and standard deviations calculated from three replicates (shown here as red circles and individually in Supplementary Fig. 4). For growth in galactose media, see Supplementary Fig. 4. For BY4742 and BY4742Δa microscope images, see Supplementary Fig. 3d, e. Scale bar = 10 μm. **$p < 0.01$, ****$p < 0.0001$.

in accordance with findings in other genetic backgrounds[29,30,32]. *AMN1* deletion causes a more robust, irreversible transition to unicellularity compared to *AMN1* mutations that arose during experimental evolution[27], suggesting the latter may be weaker or partially reversible. In various standard growth media *AMN1* deletion speeds up growth either through beneficial effects of unicellularity or by pleiotropically elevating uptake and conversion of sugar into biomass.

**Loss of *AMN1* impairs TBR1 growth in stressful conditions.** Considering the tradeoff between normal growth and stress resistance[27,28,52], we asked if the TBR1 strain is more drug resistant than TBR1Δa, either due to multicellularity or other effects of *AMN1*. To address this question, we compared the growth curves of TBR1 and TBR1Δa in normal conditions to their growth curves in increasing concentrations of four chemical stressors: the oxidative agent hydrogen peroxide ($H_2O_2$) and drugs representing the three main classes of antifungals: amphotericin B (AmB, a polyene), caspofungin (CASP, an echinocandin), and fluconazole (FLC, an azole). We sought to understand how clumping or *AMN1* loss affect response to

treatment in two ways: first, by analyzing entire growth curves globally and then, by estimating specific local parameters corresponding to various growth phases.

To globally characterize entire growth curves and their drug-dependent differences, we calculated the area under each curve (AUC)[53] relative to the starting cell density (Fig. 2a–d, Supplementary Fig. 7a–d), a fitness measure that estimates the cumulative lifespan of all cells in the sample. AUC[54] quantifies the total time-duration obtained by piecing together all cell cycle times throughout the growth curve, and subtracting all time after cell death. The AUC of unicellular TBR1Δa cells decreased compared to the parental TBR1 strain in all four types of stress (Fig. 2e, Methods). Nonetheless, the shaded areas in Fig. 2a–d indicated that various stresses lower the TBR1Δa strain's AUC differently, by reshaping the growth curves stress-specifically compared to the growth curve in unstressed condition. Here, we define growth curve reshaping as changes in the number, slope and duration of growth phases that cause a drug-induced drop in the AUC compared to the stress-free conditions.

To gain local insights into stress-specific growth curve reshaping, we plotted the $OD_{600}$ absorbance values over time

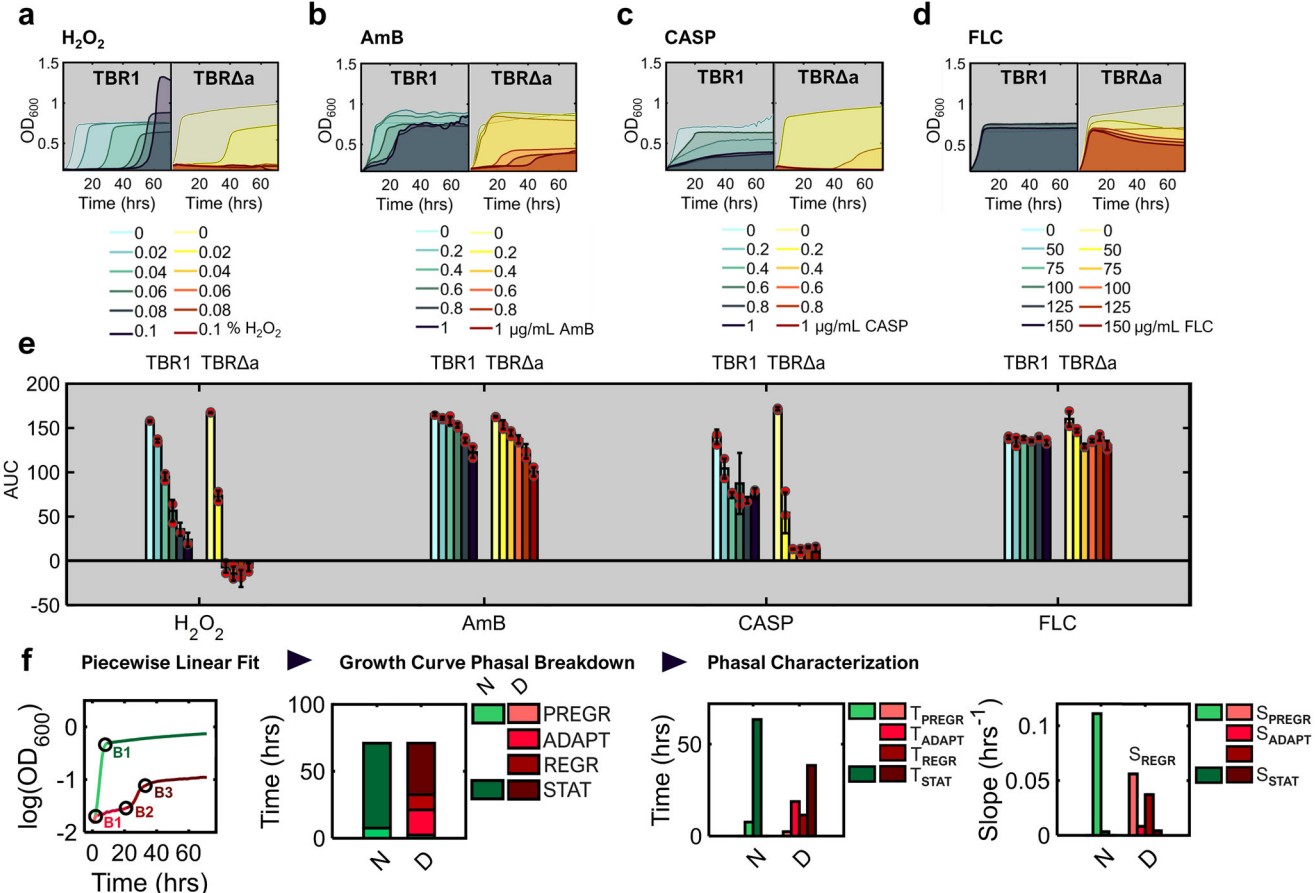

**Fig. 2 AUC representing total cumulative lifespan, and growth curve reshaping in TBR1Δa strain versus the parental TBR1 strain. a–d** Drug-dependent shrinkage of the shaded area under growth curves ($OD_{600}$) upon exposure to (**a**) hydrogen peroxide, $H_2O_2$; (**b**) amphotericin B, AmB; (**c**) caspofungin, CASP; and (**d**) fluconazole, FLC. Here, the representative replicates are shown. For all replicates, see Supplementary Fig. 7a–d. **e** Area under each growth curve (AUC) above starting population size, approximated by numerical integration via the trapezoid method with equally spaced 1-h intervals. Red circles represent individual data points. Error bars represent means and standard deviations calculated from AUC of three biological replicates. **f** Growth curve analysis by piecewise linear fits to ln($OD_{600}$) versus time is exemplified by TBR1Δa in normal (N) and 0.8 μg/ml AmB drug-containing (D) medium. The circles and letters next to them (B1, B2, B3) indicate breakpoints identified by the piecewise linear fitting within each growth curve. The breakpoints divide the N curve into 2, and the D curve – into 4 phases: pregrowth, adaptation, regrowth, and stationary phase. To characterize growth curve reshaping, the slope (S) and duration (T) of each growth phase (Supplementary Fig. 7e–h) were calculated for all drug concentrations.

on the semilogarithmic scale. The resulting growth curves became approximately piecewise linear. Next, we applied a piecewise linear fitting algorithm (Methods) to identify the coordinates of breakpoints that separate quasilinear segments (i.e., growth phases based on at least 10% slope change, Fig. 2f and Supplementary Fig. 7e–h) within each growth curve. The breakpoint coordinates thus define the duration and slope of each segment, providing a detailed characterization of stress-dependent growth curve reshaping before reaching the final cell density. Overall, stress-induced reshaping generated two to four of the following growth phases, each characterized by its duration and slope: (1) pregrowth, (2) adaptation, (3) regrowth and (4) stationary phase. Next, we describe how each stress affects various growth curve phases of the model unicellular and clump-forming yeast strains (Supplementary Fig. 8).

**AMN1 deletion sensitizes TBR1 yeast to hydrogen peroxide.** Oxidative agents such as $H_2O_2$ have a wide fungistatic effect on both uni- and multicellular yeasts[55,56] due to lipid peroxidation, oxidation of proteins and DNA lesions[57] (Fig. 3a). Whereas the MIC of $H_2O_2$ (the lowest concentrations preventing growth) for unicellular *S. cerevisiae* strains is 4 mM[55] or ≈0.01% in liquid

solution, surface-attached yeast biofilms are more resistant to this oxidative agent[58]. Here, we studied if *AMN1* deletion may similarly sensitize TBR1 yeast to $H_2O_2$ in solution by abrogating multicellular clumping or by other pleiotropic effects.

To compare the sensitivity of clumping TBR1 and its unicellular derivative TBR1Δa to hydrogen peroxide, we analyzed growth curve reshaping by piecewise linear fitting for three replicate cultures, each exposed to $H_2O_2$ doses increasing from 0% to 0.1% (Methods) for 72 hours without resuspension. This analysis generated two to four growth phases and corresponding breakpoints, defining pregrowth, adaptation, regrowth, and stationary phases (Fig. 3b, Supplementary Figs. 7e and 9). Since pregrowth was always very short, we focused on the remaining one to three growth phases for TBR1 and TBR1Δa.

The duration of the adaptation phase from these fits lengthened with the $H_2O_2$ concentration in both strains (Fig. 3c). After adaptation, TBR1 cells regrew within the timespan of the experiment (72 hours) at rates that were not strongly stress-dependent, diminishing compared to the control at most by 1.4-fold, and correspondingly lengthening the regrowth phase durations (Supplementary Fig. 9b). The average stationary phase $OD_{600}$ value (carrying capacity) increased in two out of three TBR1 replicates at the highest $H_2O_2$ concentration (Fig. 3d).

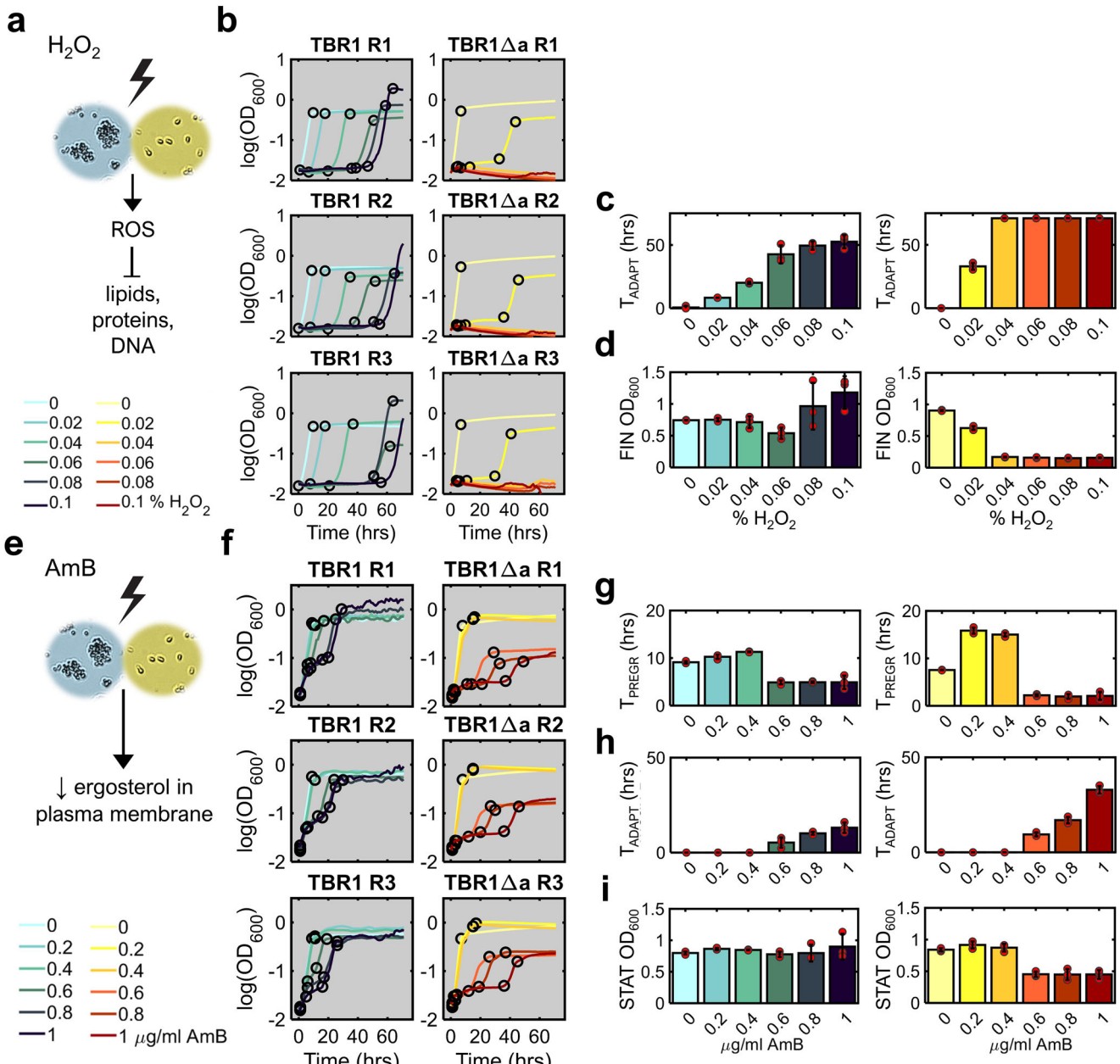

**Fig. 3 TBR1 and TBR1Δa growth curve reshaping by increasing concentrations of hydrogen peroxide and amphotericin B. a** Schematic illustration of the predicted $H_2O_2$ effect on yeast cells. **b** Growth kinetics of TBR1 and TBR1Δa exposed to increasing $H_2O_2$ concentrations, shown as $\log(OD_{600})$ over 72 hours. Black circles indicate the breakpoints identified by piecewise linear fitting. **c** Adaptation phase duration ($T_{ADAPT}$) versus $H_2O_2$ concentrations for TBR1 and TBR1Δa. **d** Mean $OD_{600}$ value in final phase (FIN $OD_{600}$) for TBR1 and TBR1Δa. Plotted values are stationary phase averages for TBR1 and average $OD_{600}$ values over the entire time course (72 hours) for TBR1Δa if regrowth does not occur. **e** Schematic illustration of the predicted AmB effect on yeast cells. **f** Growth kinetics of TBR1 and TBR1Δa exposed to increasing AmB concentrations, shown as $\log(OD_{600})$ over 72 hours. Black circles indicate breakpoints identified by piecewise linear fitting. **g** Pregrowth phase duration ($T_{PREGR}$) for TBR1 (blue) and TBR1Δa (yellow). **h** Adaptation phase duration ($T_{ADAPT}$) for TBR1 (blue) and TBR1Δa (yellow). **i** Carrying capacity (stationary phase mean $OD_{600}$ values) for TBR1 (blue) and TBR1Δa (yellow). Red circles represent individual data points. Error bars represent means and standard deviations calculated from three biological replicates shown in panel (**b**) (for $H_2O_2$) and (**f**) (for AmB).

In contrast to TBR1, $H_2O_2$ stress affected TBR1Δa growth curves more severely. The lowest $H_2O_2$ concentration (0.02%) prolonged the adaptation phase and reduced the carrying capacity (stationary phase absorbance value) compared to the no-stress control. Higher $H_2O_2$ concentrations prevented regrowth within 72 hours, with decreasing $OD_{600}$ values and negative slopes indicating cell death (Fig. 3d, Supplementary Fig. 9b). Accordingly, we detected a single breakpoint and no

TBR1Δa regrowth or stationary phase above 0.04% $H_2O_2$ (Fig. 3c).

Overall, these results are consistent with previous MIC measurements, but reveal in detail how TBR1 and its TBR1Δa derivative strain responded differently to $H_2O_2$ treatment. *AMN1* deletion sensitized the TBR1Δa strain to $H_2O_2$, causing it to adapt slower and less robustly than the parental TBR1 strain. The TBR1Δa growth curve in the lowest $H_2O_2$ concentration (0.02%)

closely resembled that of TBR1 in 0.06% $H_2O_2$, supporting the dose-dependent downshift in resistance due to *AMN1* deletion.

**AMN1 deletion sensitizes TBR1 yeast to Amphotericin B**. Next, we characterized response to Amphotericin B (AmB), a polyene antifungal that can traverse the cell wall and bind to ergosterol, rapidly disrupting yeast cell membranes (Fig. 3e)[59] with a typical $MIC_{90}$ of 1 μg/mL[60]. While considered fungicidal, AmB could also exhibit static activity at sub-$MIC_{90}$ doses[61]. To characterize growth response to AmB, we analyzed the TBR1 and TBR1Δa growth curves at 0, 0.2, 0.4, 0.6, 0.8, and 1 μg/mL doses of AmB in YPD for 72 hours (Methods).

Based on piecewise linear fitting, growth curve responses to AmB were less drastic and less different than in $H_2O_2$. Both TBR1 and TBR1Δa cells showed a somewhat concentration-dependent, delayed response to the addition of AmB, followed by adaptation, regrowth, and stationary phase. Whereas the lowest AmB concentrations (0.2, 0.4 μg/mL) did not reshape the growth curve relative to the YPD control (Fig. 3f, Supplementary Figs. 7f and 10a), the three highest drug concentrations suppressed the growth of both strains (Supplementary Fig. 10c), an effect that lasted longer for unicellular TBR1Δa cells (Fig. 3h). Both strains regrew slower than without drug (Supplementary Fig. 10b–d), their regrowth slopes dropping 5.75-fold and 2.06-fold compared to 0 μg/mL AmB for TBR1Δa and TBR1 cells, respectively. Finally, the carrying capacity was unaltered for TBR1, but was approximately halved for TBR1Δa (Fig. 3i). The pregrowth phase was generally steeper in TBR1 than TBR1Δa (Supplementary Fig. 10a, b) and lasted longer (Fig. 3g), which illustrates higher initial tolerance to AmB. Overall, piecewise linear fitting indicated that AMN1 deletion sensitized yeast to AmB, which manifests in prolonged adaptation and lower carrying capacity.

**AMN1 deletion sensitizes TBR1 yeast to Caspofungin**. Caspofungin (CASP) is a representative of echinocandin antifungals that prevents the synthesis of an essential cell wall component by blocking the enzyme β-1,3-D-glucan synthase (Fig. 4a)[62], with a typical $MIC_{90}$ of 0.12–1 μg/mL[63,64].

To investigate how caspofungin reshaped the TBR1 and TBR1Δa growth curves, we exposed both strains to 0, 0.2, 0.4, 0.6, 0.8, and 1 μg/mL of CASP in YPD for 72 hours (Methods). CASP reshaped TBR1 growth curves into three phases that were less distinguishable than for other stresses (Fig. 4b, Supplementary Figs. 7g and 11a). The pregrowth phase was generally short (Supplementary Fig. 11b). The adaptation and regrowth phases merged into a single phase for TBR1, but not TBR1Δa. The regrowth phase had CASP concentration-dependent, reduced steepness compared to the control (Fig. 4c). The carrying capacity decreased with the drug concentration (Fig. 4e). In contrast, almost all drug concentrations killed the TBR1Δa strain after a short pregrowth phase, the duration of which dropped to 1 hour at all concentrations above 0.4 μg/mL (Supplementary Fig. 11b). Therefore, only two growth phases were typically present for TBR1Δa, which regrew only at the lowest CASP dose (0.2 μg/mL), but almost 50 hours after inoculation (Supplementary Fig. 11c). Thus, we considered TBR1Δa adaptation time equal to the time span of the experiment (72 hours) for all higher CASP concentrations (Fig. 4d).

**AMN1 deletion sensitizes TBR1 yeast to Fluconazole**. Fluconazole is a frequently used azole that targets a fungal cytochrome P450 enzyme (lanosterol 14-α-demethylase) thereby inhibiting the synthesis of ergosterol (Fig. 4f)[65]. Accounting for the resistance threshold value (64 μg/mL)[66] and the static and cidal effects observed below and above this value[67], we characterized growth

curve reshaping by inoculating TBR1 and TBR1Δa in 0, 50, 75, 100, 125, and 150 μg/mL solutions of FLC in YPD. As opposed to the other three stressors, none of the tested FLC concentrations altered the growth curve before stationary phase (Fig. 4g, Supplementary Fig. 7h), leaving the exponential growth rates practically unaffected. Thus, all TBR1 and TBR1Δa growth curves had two (pregrowth and stationary) phases and a single breakpoint at ~10 hours after inoculation. Interestingly, while stationary phase absorbance remained stable for TBR1, it started declining for TBR1Δa with a drug concentration-dependent negative slope (Fig. 4h). This suggested that FLC kills TBR1Δa cells with a substantial delay surpassing the time to stationary phase, and that resuspending TBR1 and TBR1Δa cells in FLC before they reach stationary phase might sensitize exponential phase cells to FLC.

To test this hypothesis, we resuspended the cells in fresh media with corresponding FLC doses after 10 hours of incubation with FLC (Supplementary Figs. 12 and 13) and continued culturing them for another 72 hours. The reshaped growth curves (Fig. 4i, j, Supplementary Figs. 12a–c and 13d) were reminiscent of those resulting from AmB exposure (Supplementary Fig. 13) confirming delayed FLC action. Interestingly, TBR1 growth curves were also affected upon resuspension, indicating that, given sufficient time, FLC affects actively growing cells much more than stationary phase cells.

Overall, the chosen range of FLC concentrations did not affect the TBR1 strain without resuspension, while it affected the TBR1Δa strain only in stationary phase, consistent with the documented delay of FLC action[68]. The FLC concentration-dependent stationary phase slope decrease indicates cell death independent of sugar uptake. Resuspension in FLC before stationary phase demonstrated that FLC acts late, especially on actively growing cells in a manner resembling AmB, as expected since both drugs affect the ergosterol synthetic pathway, although with different delays.

**AMN1 deletion tends to enhance drug resistance of non-clumping yeast**. After observing more severe stress-dependent growth curve changes in unicellular TBR1Δa compared to clumping TBR1, we sought to investigate the effect of *AMN1* deletion independent of clumping. We therefore exposed two unicellular S288 lab strains, BY4742 and BY4742Δa, as well as the mostly unicellular TBR1 EvoTop (Supplementary Figs. 15 and 16, Supplementary Note 4) to the same drugs at identical doses (0, 0.02, 0.04, 0.06, 0.08, 0.1 % $H_2O_2$; 0, 0.2, 0.4, 0.6, 0.8, 1 μg/mL AmB; 0, 0.2, 0.4, 0.6, 0.8, 1 μg/mL CASP; 50, 75, 100, 125, 150 μg/mL FLC) following the same experimental setup (Methods).

The drugs reshaped BY4742 and BY4742Δa growth curves into growth phase patterns similar to TBR1 and TBR1Δa, implying drug- rather than strain-specific growth curve alteration (Fig. 5a–d). However, contrary to TBR1 and TBR1Δa, BY4742 appeared more sensitive to both $H_2O_2$ and AmB than its *AMN1*-deficient BY4742Δa derivative (Fig. 5e, f, respectively). *AMN1* loss sensitized BY4742 cells only to CASP (Fig. 5c, h), whereas FLC lowered stationary-phase cell counts similarly for both unicellular strains (Fig. 5d, g). As for TBR1 and TBR1Δa, the effect of FLC increased similarly for BY4742 and BY4742Δa resuspended before reaching stationary phase.

Overall, these results indicated that *AMN1* loss enhances rather than weakens resistance of non-clumping laboratory yeast to multiple stressors, although not to CASP. Thus, the presence of *AMN1* tends to sensitize unicellular BY4742 to drugs, as opposed to the higher resistance that *AMN1* confers to multicellular TBR1. This effect reversion suggests that clumping could enhance resistance in TBR1 versus TBR1Δa, offsetting other pleiotropic effects of *AMN1*. However, the genetic background of standard

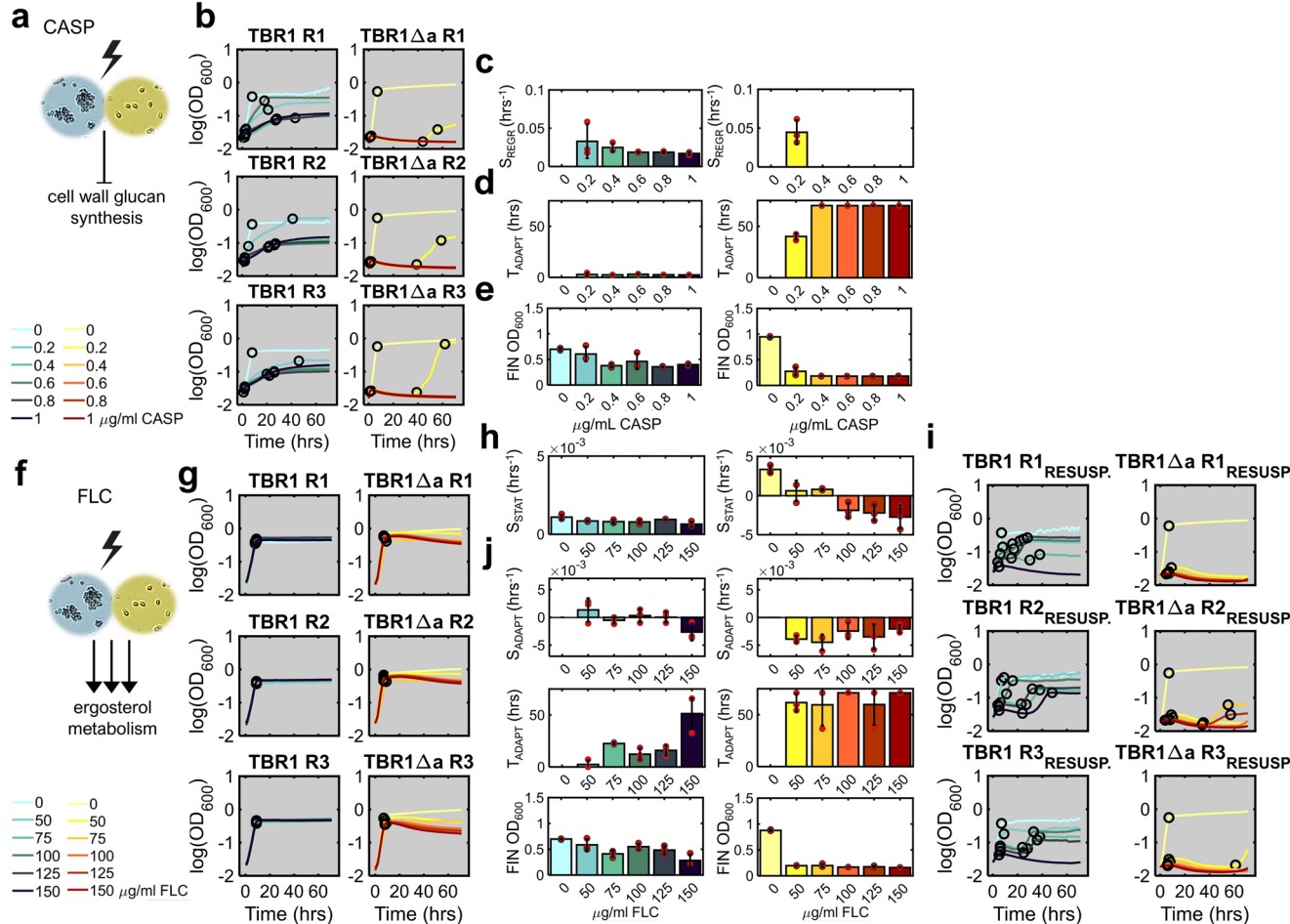

**Fig. 4 TBR1 and TBR1Δa growth curve reshaping by increasing concentrations of caspofungin and fluconazole. a** Schematic illustration of CASP effect on yeast cells. **b** TBR1 and TBR1Δa growth kinetics in various CASP concentrations, shown as log(OD$_{600}$) over 72 hours. Black circles indicate the breakpoints identified by piecewise linear fitting. **c** The slopes of the regrowth phase (S$_{REGR}$), which for TBR1Δa only occurred at 0.2 µg/mL CASP. **d** The duration of the adaptation phase (T$_{ADAPT}$). **e** The average final phase OD$_{600}$ values (FIN OD$_{600}$). **f** Schematic illustration of FLC effect on yeast cells. **g** TBR1 and TBR1Δa growth kinetics in various FLC concentrations, shown as log(OD$_{600}$) over 72 hours without resuspension. Black circles indicate breakpoints identified by piecewise linear fitting. **h** The stationary phase slopes (S$_{STAT}$) for TBR1 and TBR1Δa without resuspension. **i** TBR1 and TBR1Δa growth kinetics shown as log(OD$_{600}$) with resuspension into the same FLC concentrations before stationary phase. Black circles indicate the breakpoints identified by piecewise linear fitting. **j** The slope and duration of the adaptation phase (S$_{ADAPT}$, T$_{ADAPT}$), as well as the final phase mean OD$_{600}$ value (FIN OD$_{600}$) for resuspended cells. Red circles represent individual data points. Error bars represent means and standard deviations calculated from three biological replicates shown in panel **b** (for CASP), **g** (for FLC without resuspension), and **i** (for FLC with resuspension).

laboratory BY4742 strains[69] also deviates substantially from TBR1[70], and an Asp368Val polymorphism in the Amn1 sequence of TBR1 versus BY4742 may alter Amn1 function. Disentangling the effects of these genetic differences from those of clumping multicellularity warrants future studies.

**Mathematical model captures drug-specific growth curve reshaping.** Inspired by recent models of aging[71] that produces harmful senescent cells[72] which the human body attempts to remove by processes that can saturate[71], we developed a generic mathematical model to capture drug-specific growth curve reshaping. Considering that yeast cells respond similarly to aging and stress[73], we hypothesized that stressed yeast cells neutralize secondary toxic chemicals (such as reactive oxygen species, ROS) like the human body tries to eliminate senescent cells. Thus we expanded the sugar utilization models (Supplementary Note 2) by adding ordinary differential equations (ODEs) to account for drug influx and degradation/dilution with growth-mediated feedback[74,75], toxicity accumulation proportional to the intracellular drug concentration, cellular growth inhibition and killing,

and cellular detox[76,77] as a saturating enzymatic step[71] (Fig. 6a, Supplementary Table 5, Supplementary Note 3). We used nonlinear least-squares optimization to fit these models (Supplementary Figs. 17–20) to cell count estimates (Supplementary Note 1, Supplementary Fig. 5) obtained from experimental OD$_{600}$ absorbance data (Fig. 6c). For each drug, we gradually constrained parameters, such as the spontaneous drug degradation rate that should not be drug concentration or strain dependent. The AUC estimated from these model fits matched closely the experimental AUC values (Fig. 6d, Supplementary Figs. 21 and 22), indicating that the same model could capture all growth curve changes for all stressors for both TBR1 and TBR1Δa.

To understand mechanistic differences between various stressors, and to uncover how *AMN1* deletion might sensitize TBR1 to drug treatment (Fig. 6b) we studied how various parameters changed upon ODE solution fitting to TBR1 and TBR1Δa data (Supplementary Note 3). Whereas the best-fit parameters tended to be similar across all concentrations of the same drug, they changed substantially between different drugs. Although we did not include clumping explicitly in the model, the

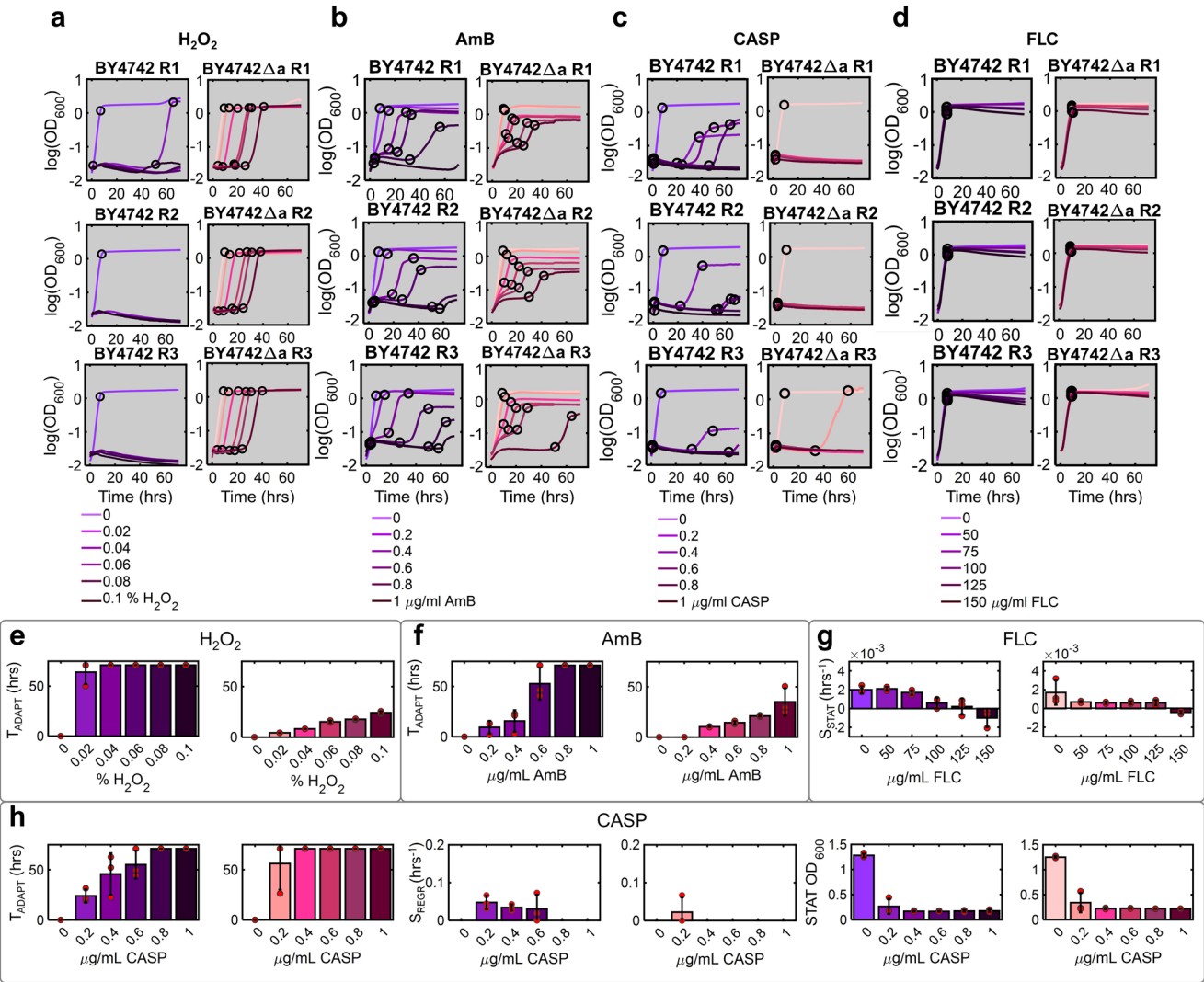

**Fig. 5 BY4742 and BY4742Δa growth curve reshaping by increasing concentrations of antifungals. a–d** BY4742 (shades of purple) and BY4742Δa (shades of magenta) growth kinetics in various **a** $H_2O_2$, **b** AmB, **c** CASP, and **d** FLC concentrations, shown as $\log(OD_{600})$ over 72 hours. Black circles indicate the breakpoints identified by piecewise linear fitting. **e** Adaptation duration ($T_{ADAPT}$) affected by $H_2O_2$ exposure. **f** Adaptation duration ($T_{ADAPT}$) affected by AmB exposure. **g** Stationary phase slope ($S_{STAT}$) affected by FLC exposure. **h** Adaptation duration ($T_{ADAPT}$), exponential phase slope ($S_{REGR}$), and the average stationary phase $OD_{600}$ value (STAT $OD_{600}$) affected by CASP exposure. All parameters are shown for BY4742 and BY4742Δa growth conditions in panels **a–d**. Red circles represent individual data points. Error bars represent mean and standard deviation calculated from three biological replicates shown in panels **a–d**.

rate $f$ of drug diffusion into/out of the cells is most likely to depend on clumping, which should lower drug influx by reducing the average effective cellular surface area exposed to the extracellular environment. Therefore, upon fitting the same model to both clumpy and unicellular data, we tried inferring the effect of clumping from changes in the parameter $f$ versus other parameters. The fits (Fig. 6c, Supplementary Fig. 21) indicated that (i) the drug in/outflux rate $f$ was generally lower in the clumpy TBR1 versus the unicellular TBR1Δa strain; but also (ii) the growth-inhibiting threshold $q$ of intracellular drug concentration tended to be lower in TBR1 versus TBR1Δa; and (iii) the detox rate $p$ tended to be higher in TBR1 versus TBR1Δa. Other parameters changed in drug-specific ways that were less consistent or even antagonistic, such as highly enhanced TBR1Δa killing offsetting lower sensitivity to FLC growth inhibition, or higher CASP influx and weaker detox offsetting reduced toxicity in TBR1Δa (Fig. 6e, Supplementary Table 5). Increasing only the cell permeability parameter $f$, the main culprit of multicellular drug resistance, could not convert TBR1 fits into TBR1Δa growth

curves for any drug, indicating that drug sensitization by *AMN1* deletion involves other complex mechanisms besides elevated drug influx. Overall, this suggests that Amn1 affects resistance in a variety of ways, and clumping is only one of the multiple resistance mechanisms that further studies will need to disentangle.

The FLC growth curves required introducing an extra parameter $b$ to capture the death of actively sugar-consuming, growing cells, in addition to the sugar-independent, general killing rate $k$. First, capturing that FLC did not affect TBR1, but killed TBR1Δa in sugar-depleted stationary phase required a similar, modest $k$ for TBR1 versus TBR1Δa. Yet, incorporating such modest $k$ values could not capture actively growing TBR1 cells, and even less TBR1Δa being killed upon resuspension.

We investigated similarly, by experiment and modeling, the effect of TBR1 and TBR1Δa incubation with sporadic mild versus continuous intense orbital shaking[78] (Methods) and found minimal differences in most conditions, except in FLC, which

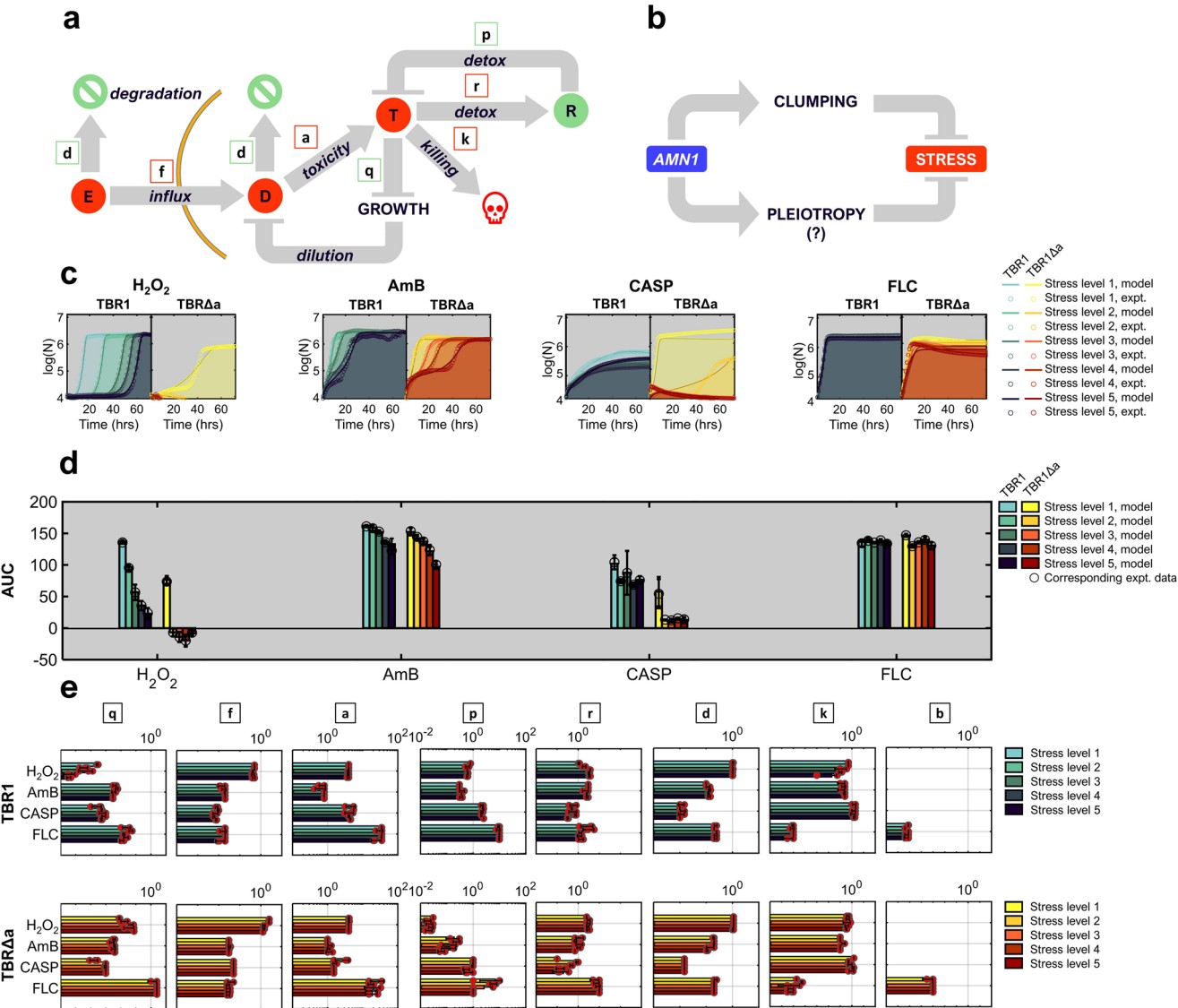

**Fig. 6 Mathematical model captures drug-drug and strain-strain differences. a** Schematic illustration of modeling the variables E (external drug), D (internal drug), T (toxicity), R (detox machinery). Green-colored species have a positive effect on growth, whereas the red-colored ones suppress growth. **b** Schematic illustration of how *AMN1* may promote stress resistance. **c** Simulated growth curves (solid lines) co-plotted on the logarithmic scale with representative experimental cell count-converted curves (dotted lines) with the shaded area illustrating the area under the curve, AUC. For all replicates, see Supplementary Fig. 21. **d** The AUC calculated from simulated growth curves in panel **c** shown as bars, compared to AUCs of the experimental cell count-converted data shown as scatter dots. Error bars represent means and standard deviations calculated from fits to three experimental replicates. Black circles represent corresponding experimental AUC data points. **e** Parameters for growth curve models: q – internal drug threshold to inhibit growth, f – drug in/outflux rate, a – drug-induced cell toxicity, p – detox production rate, r – drug threshold to induce detox, d – spontaneous drug decay, k, b – cell killing/death rate (Supplementary Table 5, Supplementary Note 3). For the entire model output, see Supplementary Figs. 17–20. Bars are shown for TBR1 (shades of blue) and TBR1Δa (shades of yellow) growth models. Color gradient from brighter to darker shades represents the increasing stress levels of H₂O₂, AmB, CASP, and FLC. Red circles represent the values of each parameter per fit to an individual experimental replicate. Error bars represent means and standard deviations calculated from the ODE models of three experimental biological replicates.

affected the exponential phase of both intensely shaken strains (Supplementary Fig. 23).

To similarly investigate the effect of *AMN1* deletion on the drug-dependent growth of a unicellular laboratory strain, we fit the same model to all BY4742 and BY4742Δa (Supplementary Fig. 24) growth curves. First, we focused on drug influx rate, *f*. If clumping lowers drug influx, as the models suggested in TBR1, then it should not play a role in unicellular BY4742. Indeed, the drug influx parameter *f* did not change considerably in BY4742Δa versus BY4742. Likewise, most other parameters were similar in BY4742Δa versus BY4742, except for the lower

threshold (*r*) for H₂O₂ and AmB detox activation in BY4742Δa versus BY4742, which might explain the increased BY4742Δa resistance to these drugs. Interestingly, similar to the TBR1 strains, detox (*p*) was lower for all drugs in BY4742Δa, possibly explaining higher sensitivity to CASP (Supplementary Fig. 24).

Overall, these findings suggest that higher drug resistance in TBR1 yeast cells is due to both pleiotropic and morphological effects of Amn1 (i.e., multicellularity). Besides abrogating multicellularity and elevating drug influx in TBR1, *AMN1* deletion might elevate or reduce stress sensitivity in various strains

through unknown general detox and other drug-specific mechanisms that remain to be investigated.

## Discussion

Multicellularity is a drug resistance mechanism that typically limits drug influx in fungi[25], bacteria[6,79], and mammalian cells[80,81]. Nonetheless, it remains unclear if stress resistance stems directly from multicellularity or from pleiotropic effects of mutations conferring multicellularity. Many antifungals used in the clinic (e.g., ergosterol-targeting drugs such as AmB and FLC) have dose- and strain-dependent variation of static and cidal effects[61,82], which complicates the interpretation of cellular resistance. Detailed, quantitative investigation of drug effects and their mechanisms would standardize their usage in the clinic and laboratory research. Current drug resistance quantification methods are still lacking sophistication to quantitatively pinpoint drug-specific differences among stressors and strains. To describe AMN1's effects on multicellularity and molecular pleiotropy as possible drug resistance mechanisms in baker's yeast, we developed quantitative analysis of global and local differences between growth curves. The AUC relative to the starting cell concentration[54] indicated that AMN1 deletion sensitized TBR1 yeast to all stressors. Piecewise linear fitting separated each growth curve into distinct growth phases with slopes and durations implying that: (i) hydrogen peroxide acts quickly but provokes adaptive response that is weaker in TBR1Δa; ii) amphotericin B reshapes growth curves with a slight delay but the cells can adapt and regrow, although later in TBR1Δa; iii) caspofungin compresses or flattens growth curves, especially in TBR1Δa; and lastly iv) fluconazole affects only the stationary phase of TBR1Δa, but if cells are resuspended then fluconazole hits both strains similarly to amphotericin B, potentially due to related mechanisms of action, although more severely for TBR1Δa. Overall, the TBR1Δa strain, genetically identical to the clumping TBR1 strain, except for AMN1 deletion, is more sensitive, in specific ways to all stressors. In contrast to TBR1, AMN1 deletion tended to sensitize the non-clumping, unicellular lab strain BY4742 to stressors except caspofungin. This suggests that AMN1-conferred clumping and detox may enhance resistance in TBR1 by offsetting other, drug-sensitizing effects of AMN1.

To understand the mechanisms of stress-sensitization by AMN1 deletion, we sought to capture experimental growth curve reshaping via an ODE model accounting for drug influx and degradation, drug toxicity accumulation, toxicity-induced growth inhibition and death, and cellular detoxifying response. Similar models could be applicable to elucidate mechanisms of stress-dependent growth curve reshaping for other microbes or even cancer cells. The same model captured experimental responses to all four agents at all concentrations in all strains, and data-fitting suggested parameters responsible for growth curve differences among stressors and strains. Although trapping in local optima is often possible with multi-parameter estimation, we found that the parameters generally capturing AMN1 deletion effects were drug influx, intracellular sensitivity and detox rate. Thus, drug penetration in clumping TBR1 is not the only effector of drug resistance, implicating other pleiotropic effects that alter drug sensitivity upon AMN1 deletion. For example, recent evidence of Amn1 ubiquitinase function[29] may suggest a role for proteasomal degradation. AMN1 deletion may prevent Ace2 degradation[29], which can misbalance cellular protein homeostasis implicating chaperone hubs such as Hsp70 and Hsp90[83–85], a major contributor to fungal drug resistance[86,87] and morphogenesis[85]. Another interesting connection could be through the Mck1 kinase involved in both stress resistance[88] and daughter cell separation[89]. Mck1 interacts directly with Ace2[90,91], which participates in a negative feedback loop with Amn1[29].

In mathematical models we assumed drug resistance by cellular sensing and stress response[76,77], without drug resistance mutations, for multiple reasons. First, resistance manifested as regrowth after reproducibly stress-correlated adaptation times, without the randomness of mutational resistance. Second, adaptation within hours is closer to the time scale of intrinsic stress response[76,77] and hardly sufficient for new mutations to fix. Other models involving mutations and persister cells will be interesting to develop and test on longer-term experimental data as before in mammalian cells[92]. Also, the molecular mechanisms of altered detoxification suggested by the model will be interesting to pursue.

Based on these findings, AMN1 could be a potential target for inhibitors that could sensitize clumping fungal pathogens not only to common antifungals, but also to immune cell attack[93]. Quantitative growth curve analysis and modeling combined with genetic perturbations could identify other AMN1-like genes that alter drug resistance strain-specifically through multicellularity and other effects. While it remains to be determined how growth curve parameters in suspension relate to drug effects on surface-attached yeast, these methods could promote the development or repurposing of drugs against fungal pathogens, addressing the major current medical challenge of antifungal[37,38], and more broadly, antimicrobial[1] resistance. In summary, the parameters and methods we introduce should be important for predicting the responses of various uni- and multicellular microbes and even cancer cells to various stressors, including emerging antifungals and other therapeutic chemicals.

## Methods

**Yeast strains and growth media**. In this work we used the *Saccharomyces cerevisiae* strain TBR1 (Σ1278b strain 10560-23C; MATα, ura3-52, his3::hisG, leu2::-hisG), obtained in previous studies by multiple crosses of baking strains Yeast Foam and 14211D[27,70] that exhibits deficient mitotic cell separation phenotype, forming three-dimensional clumps. Distantly related to the 'classical' lab strain S288c, TBR1 carries 44 unique genes and 3.2 single-nucleotide polymorphisms per kilobase compared to the standard sequence[70]. Experimental evolution of the TBR1 strain with selection against fast-sedimenting clusters led to transition to mostly unicellular phenotype with decreased stress tolerance[27]. High-throughput whole-genome sequencing revealed that the evolved unicellular EvoTop TBR1 strain differed from the ancestral clump-forming strain mainly by mutations in the *AMN1* coding sequence. We also used *S. cerevisiae* KV38 and YPH500 as clumping and non-clumping controls, respectively. BY4742[69] (MATα, his3Δ1, leu2Δ0, lys2Δ0, ura3Δ0) and BY4742Δa (Horizon YSC6272-201919655) were used as standard laboratory strains for drug response comparisons.

Prior to imaging, absorbance measurements and other procedures, yeast cultures were maintained in a LabNet I5311-DS shaking incubator at 30 °C, 300 rpm with OD$_{600}$ absorbance measurements taken once per hour. In all experiments, 10$^5$ cells (approx. 0.05 < OD$_{600}$ < 0.1) were inoculated into yeast peptone dextrose (YPD) liquid growth medium containing Yeast Extract (MilliporeSigma Y1625) 1% w/v, Peptone (BD Bacto™ 211677) 2% w/v, D-Glucose (MilliporeSigma G7528) 2% (w/v). Medium for *kanMX* selection in TBR1Δa was YPD containing 200 μg/ml geneticin (G418)[94]. Spectrophotometry before microscopy was performed using a Unico System S-1205 spectrophotometer with calibration against the corresponding blank medium. In all time-course experiments, optical density (OD$_{600}$) absorbance measurements were taken hourly using Tecan Infinite® 200 PRO microplate reader in Falcon® 96-well flat-bottom plates (Corning 351172), one replicate per well (total 200 μl/well), in total three replicates. The incubation in the microplate reader over the course of 72 hours was interrupted by 1 minute of orbital shaking for every 9 minutes of steady incubation (for mild periodic shaking mode) or consisted of uninterrupted ≈45 minutes shaking and ≈15 minutes steady incubation (for continuous intense shaking mode).

**Plasmid construction**. The *AMN1* knock-out plasmid was based on homology recombination[47] within a specific genomic locus (*AMN1*) of TBR1 strain (Supplementary Fig. 1) and was assembled following the NEB® HiFi Assembly Protocol for *E. coli* NEB® 10-beta competent cells. Plasmid DNA extraction procedures were carried out using correspondent QIAGEN® protocols. Yeast genomic integration was confirmed by Sanger Sequencing of the colony PCR product of correspondent genomic locus at the Stony Brook DNA Sequencing Facility (Supplementary Fig. 2). Yeast transformation and yeast genomic DNA extraction were performed using EZ-Yeast™ Transformation Kit (MP Biomedicals™) starting from liquid culture and MasterPure™ Yeast DNA Purification Kit (Thermo Fisher Scientific NC9756781), respectively, following the Manufacturers' protocols. Primers used in this study are listed in Supplementary Table 1.

**Microscopy and image processing**. Image acquisition was preformed using Cellometer® Vision CBA Image Cytometer (Nexcelom Bioscience LLC.). For image analysis we used Nexcelom Data Package and MATLAB software. Nexcelom filter VB-595-502 was used for red fluorescence assay in Cellometer. We used Nexcelom image segmentation (Supplementary Figs. 3 and 8) for object detection upon setting Cell Diameter parameter to 2–5 μm for single cells and 5–20 μm for clumps[95]. The Object Roundness parameter was set to 0.45-0.8 (with 1.0 used for perfectly circular shapes) to separate single living cells from debris.

**Stress and drug resistance assays**. The environmental stress factor hydrogen peroxide ($H_2O_2$) was used to elicit general stress response, whereas antifungal agents, amphotericin B (AmB), caspofungin (CASP), and fluconazole (FLC), were used as specific, clinically relevant antifungals. AmB (Thermo Fisher Scientific 15290-018) was diluted in liquid YPD and YPD + G418 medium (for TBR1 and TBR1Δa, respectively) and added to growth medium in fungistatic concentrations[96] 0.2, 0.4, 0.6, 0.8, and 1 μg/ml. CASP (Cayman Chemical 15923) stock solution was prepared in 96% ethanol and added to YPD and YPD + G418 in concentrations in the range of $MIC_{90}$ for *C. albicans*: 0.2, 0.4, 0.6, 0.8, and 1 μg/ml[63,64]. Fluconazole (R&D Systems 3764) was diluted in distilled water and added to the growth media to final concentrations 50, 75, 100, 125, and 150 μg/ml, approximating the concentrations survived by candidiasis-inducing biofilms in clinical samples[66]. After 24 hours of growth in liquid rich medium, the cells were harvested by centrifugation and resuspended in $H_2O_2$-, AmB-, CASP-, and FLC-containing YPD and YPD + G418 media.

**Growth curve analysis**. To define the growth phases, we applied piecewise linear fitting using MATLAB. After plotting the absorbance ($OD_{600}$)-based growth curve on a semilogarithmic scale, the algorithm defines the breakpoints in the behavior each growth curve. Based on the assigned breakpoint coordinates, we can further estimate the duration and the slope of each phase, defined by the two neighboring breakpoints. The AUC (Figs. 2e, 6d, Supplementary Fig. 22) was calculated in MATLAB via trapezoid integration with unit spacing over the 71-hour period with 1-hour intervals. Cell count estimation from $OD_{600}$ values (Supplementary Note 1) was based on $OD_{600}$ measurements of twofold serial dilutions for each strain in liquid media with the Tecan Infinite® 200 PRO microplate reader, followed by imaging and cell counting of the same samples in the Cellometer® Vision CBA Image Cytometer (Nexcelom Bioscience LLC.).

**Computational modeling**. For growth curve simulations, we developed a set of ordinary differential equations (ODEs), first without (Supplementary Note 2), and then with drugs (Supplementary Note 3). Without drugs, a system of two ODEs captured time-dependent changes in cell number $N$ and sugar amount $S$, as cells convert sugar into biomass with an Allee effect. With drugs, we introduced three additional ODEs to model the time-dependence of extracellular drug concentration $E$, intracellular drug concentration $D$, and cellular toxicity $T$. We also introduced a first-order Hill-type inhibitory effect of toxicity on sugar-dependent growth, a first-order Hill-type activation of cellular detox, drug diffusion into cells, and spontaneous drug degradation. We integrated the ODE systems numerically via the ode45 and ode15s MATLAB solvers. These simulated growth curves were then fit to the experimental cell count estimates using the lsqnonlin nonlinear data-fitting function in MATLAB by minimizing the least squares-metric.

**Statistics and reproducibility**. We used the paired two-tailed Student's *t*-test with unequal variance to compare cell and clump sizes between the populations of interest. We used one-tailed Student's *t*-test to compare the coefficients of variation (CV) of cell/clump size distributions, as well as experimental drug growth response parameter distributions. Means within each group were analyzed as normally distributed ratio-scale data. Statistical significance was set with $\alpha = 0.05$. All *p*-value calculations and relevant parameters are listed in Supplementary Data 1.

**Reporting summary**. Further information on research design is available in the Nature Research Reporting Summary linked to this article.

## Data availability
All raw and processed datasets underlying the main figures are available on FigShare at https://figshare.com/articles/dataset/Guinn2022_Supplementary_Data/19252004. The plasmid generated in this study is available on AddGene (#80776).

## Code availability
All codes are available on GitHub at https://github.com/lesiaguinn/Guinn2022_CODES.git.

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

## Acknowledgements

We would like to thank Uri Alon for insightful comments. We thank Matthew Wu and Ann Lin for their help in optimizing experimental protocols and Annie Lin for her help with parameter scans. We thank Bettina Fries, M. Tyler Guinn, Daniel Charlebois, and all Balázsi Lab members for valuable discussions. We thank the Todd B. Reynolds, James J. Collins, and Kevin Verstrepen labs for sharing yeast strains. This work was supported by the National Institutes of Health - NIGMS MIRA Program (R35 GM122561), the NIAID (5R01AI127704) and the Laufer Center for Physical and Quantitative Biology.

## Author contributions

L.G. and G.B. conceived the project. L.G. and E.L. performed the experiments. L.G. and G.B. analyzed the data and developed computational models. L.G. and G.B. prepared the manuscript. G.B. supervised the project.

## Competing interests

The authors declare no competing interests.
