## [Peer Review File · Communications Biology]

Reviewers' comments:

Reviewer #1 (Remarks to the Author):

Drug-Dependent Growth Curve Reshaping Reveals Mechanisms of Antifungal Resistance

Lesia Guinn, Evan Lo, Gábor Balázsi

This study addresses how different chemical stressors impact the growth of planktonic yeast with a specific emphasis on the effect of multicellularity. The manuscript contains extensive experiments and mathematical modeling towards a better quantitative understanding of drug responses of yeast cells under various stressors, growth conditions and with different planktonic lifestyles (unicellular or multicellular).

The data presentation is mostly clear but could be simplified. Both experimental and computational results are technically sound. Proper statistical tests and reasonable number of biological replicates were included. They so transparently reported all their data and statistical test results in either the main paper or supp info. Overall, I think that this manuscript is suitable for being published in *Communications Biology* upon addressing some comments below, which could improve the clarity and rigor of the data analysis and presentation. Given the extensive data the authors have presented, I do not think it's necessary to conduct new experiments.

The authors first obtained a robustly unicellular strain by a single gene knockout (AMN1) of an otherwise multicellular clumping baker's yeast strain TBR1. Three different antifungals and hydrogen peroxide were used as stressors, and AMN1-deletion was generally shown to sensitize the cells against all of them in unique ways in the sense that no universal trend/effect was suggested by the work. Although their data seem to favor that AMN1-deletion exerts its sensitizing effect mostly through reducing clump formation, the work still cannot clearly rule out other potential roles of AMN1 that are independent of clump formation.

In terms of the technical details, they applied piece-wise linear fits to growth curves on the semi-log scale under various stressors which helped them to break down the whole growth dynamics into distinct meaningful parts. In return, they reported some quantitative measurements on how different drugs and doses reshape growth dynamics with or without a functional AMN1 gene. Finally, their simple ODE model helped them to argue on how different drugs exert their effects on cell growth dynamics under stress such as by modulating drug influx, intracellular drug degradation, etc. In some control experiments, they also employed a standard non-clumping lab strain BY4742 and its AMN1-deletion. Their data suggested that AMN1 might be sensitizing this strain to the drugs in some cases. Hence, they argue that when clumping (which requires AMN1 in the TBR1 strain) is possible, it can counterbalance the otherwise sensitizing effects of AMN1.

Specific comments:

1. A major takeaway I got from the paper is that AMN1-deletion mediated clumping could contribute to increased drug tolerance/resistance in a manner that can overcome the pleiotropic effect of the gene. This message makes intuitive sense and could suggest a potential benefit of multicellularity. However, I feel this message was somewhat masked by the presentation of data, which sometimes appears a bit complicated (see below). By and large, the authors might consider elaborating on this point and present the modeling analysis in a way to better illustrate this point.

The authors might consider citing work on bacteria that exhibit collective antibiotic tolerance, though through different mechanisms (e.g Meredith et al, *Science Advances* 2018).

2. My impression is that some data analysis, while thorough, might be too complicated than needed. For example, the authors spent a significant amount of time talking about using *crecment* and piece-wise growth curve fitting to quantify growth under different conditions. I honestly found this to be somewhat an overkill. The authors seemed to be using "*crecment*" and

AUC interchangeably – if so, using “AUC” alone seems sufficient and its meaning is well understood.

3. Related to point 1, I thought it could be emphasized more in the text and modeling analysis. It is not clear how the effect of clumping is incorporated in the ODE model. As it stands, the modeling analysis gives the impression of data fitting and is somewhat disconnected from the central message (see point 1).

4. Aspects of the ODE model formulation are also somewhat unclear. For example, the authors assume growth of all their strains exhibit an Allee effect. What’s the major basis for this assumption and how critical is this assumption for the central points? It appears that one rationale is the including the effect allows better fit – if this is the major rationale, it’s good to make it clearer. Does the magnitude of the Allee term depend on the clumping capability of the strains tested?

Also, there appears to be some discrepancy in how it’s modeled: in Equation 1, the density-dependent growth rate is modeled as $(N+C)$, whereas in Section 2.3 (Supp), it’s modeled as $(N-C)$. Is this intended or the result of a typo?

5. In equation 6, why is drug uptake not considered? I would expect a term proportional to the first term in Equation 4, but with an opposite sign.

6. In Fig. 6E, it appears that ‘f’ (the drug influx rate) either does not change or gets smaller with increasing H₂O₂ or AmB and becomes larger with the increases in the other two drugs in the AMN1 deletion mutant. What should we have expected a priori? Was there prior evidence that clumping affects penetration of these different stressors differently?

7. ‘Growth curve reshaping’ is a vague term and repeatedly used without a clear introduction/definition of it.

8. Line 113: It would be helpful to provide at least a brief definition of the ‘normal settings’ in the main text.

9. Authors mention a base level growth even with 0% glucose in the experiments and attribute that to intracellularly stored sugar. That makes me wonder what the authors found for the Allee threshold (C) for 0% glucose.

10. For the equations along Lines 795 and 796, why doesn’t the toxicity mediated killing of cells contribute to nutrients?

11. What would have they seen if they had tested higher concentrations of FLC? My sense is that the tested concentrations might be too low, although they are somewhat recommended by literature (as claimed by the authors).

12. FLC starts showing its “growth reshaping” effects only once after cells are resuspended in the same type of media before stationary phase. That resuspension also means reducing the cell density. So, might protection from FLC have a cell density-dependent component too in addition to an initial delay? Would it be possible to lower the initial cell density instead of resuspension in order to give cells more time under the FLC treatments tested? A related question, I see in Fig. S11 that resuspension was done after 10 hours and into fresh media containing the original concentration of the drug. Do the authors have any idea about how much drug degradation is going on during those 10 hours?

15. Figure 1A-C: it would be nice to have scale bars.

16. Figure 1 F and G: I could not figure out which data are for glucose and which data are for galactose. Probably, galactose data is only in Supp, and this caption needs to be corrected.

17. Figure 2A: legends for the strains are missing

18. Fig. 2F: Breakpoint labels are not displayed in the figure. Also, why was not a death phase considered in labeling the linear pieces of the log (OD600) curves, although it clearly appears in some of the conditions in Fig. 2A-D?

19. Fig. S15-18: In some many cases (but not all) the results show that a tiny and temporary increase in D (intracellular drug concentration) succeeded by a multiple fold higher and much longer lasting increase in toxicity. I wonder how justifiable this is.

20. Figs. S19: Please specify which data belong to which stressor. I assumed it is in the order of H2O2, AmB, CASP, and FLC from left to right.

Reviewer #2 (Remarks to the Author):

This is a very thorough and insightful investigation of how a clumping gene (AMN1) impacts adaptation to and growth in the presence of different stresses. A few suggestions would improve the quality of this manuscript:

- the authors should discuss the fact that the 2 ergosterol-targeting antifungals have different effects on fungal cells (one is static, other is cidal)
- the authors should verify that the 2 TBR1 strains investigated don't show different clumping/clustering patterns upon exposure of the different stress agents
- a quantification of clumping of the the 2 BY4742 strains (similar to that shown in Fig1A-E) should be included to strengthen the author's conclusions regarding the roles of AMN1 in different strain backgrounds

Editor:

1. Expand the Methods to include more detail about how the model was constructed, and potentially include the additional variables suggested by Referee #1.

Response: As recommended, we included the variables suggested by Referee #1 that were reasonable. We replaced the corresponding figures (Figure 6, Figures S15-S21) with the updated model results, and updated our description of model construction with additional details in the corresponding Results and Methods sections, as well as in the Supporting Information.

2. Please quantify clumping phenotypes in TBR1 and BY4742 strains, as suggested by Referee #2.

Response: We have addressed this in the revised manuscript (Figure 1) as indicated in the response to comments 2 and 3 by Referee #2.

3. If feasible, we would strongly encourage you to test higher concentrations of FLC or whether there may be a density-dependent effect of FLC-resistance (in response to Referee #1's points 11-12).

Response: We appreciate this valuable recommendation, which we have addressed in the revised manuscript, as indicated in our response to comment 11 by Referee #1 (for higher FLC concentrations) and in our response to comment 12 by Referee #1 (for cell density-dependence of FLC effect on cells).

Furthermore, we would encourage you to include additional example applications or case studies involving this mathematical model, to highlight its value as a resource for the research community.

Response: This was a useful recommendation. This mathematical model is original to this project and has not been used in other contexts yet, so we have mentioned some potential applications of the model in the Discussion section:

“Similar models could be applicable to elucidate the mechanisms of stress-dependent growth curve reshaping for other microbes or even cancer cells.”

Reviewer #1, Specific comments:

1. A major takeaway I got from the paper is that AMN1-deletion mediated clumping could contribute to increased drug tolerance/resistance in a manner that can overcome the pleiotropic effect of the gene. This message makes intuitive sense and could suggest a potential benefit of multicellularity. However, I feel this message was somewhat masked by the presentation of data, which sometimes appears a bit complicated (see below). By and large, the authors might consider elaborating on this point and present the modeling analysis in a way to better illustrate this point.

Response: Thank you for pointing this out. We wish we could have a clearer message on how AMN1 affects drug resistance through clumping and pleiotropic effects. If AMN1 deletion would improve resistance of the unicellular lab strain BY4742 to all drugs, we could indeed argue generally that loss of clumping in TBR1 reverses this effect for all drugs. However, AMN1 deletion increases sensitivity to Caspofungin in both the lab strain and TBR1 strain, so AMN1's pleiotropic effect for Caspofungin is not reversed by the loss of clumping. Moreover, the AMN1 sequences and the genetic backgrounds of the two strains are not identical, which further complicates the picture. Despite these complexities, we tried to simplify the message while also expanding the description of parameter selection, to better reflect the dual effects of AMN1.

The authors might consider citing work on bacteria that exhibit collective antibiotic tolerance, though through different mechanisms (e.g Meredith et al, Science Advances 2018).

Response: Thank you for mentioning this interesting and relevant paper. We are citing it in the revised version.

2. My impression is that some data analysis, while thorough, might be too complicated than needed. For example, the authors spent a significant amount of time talking about using crescement and piece-wise growth curve fitting to quantify growth under different conditions. I honestly found this to be somewhat an overkill. The authors seemed be using “crescement” and AUC interchangeably – if so, using “AUC” alone seems sufficient and its meaning is well understood.

Response: Thank you for suggesting this simplification. We have replaced “crescement” with AUC throughout the manuscript.

3. Related to point 1, I thought it could be emphasized more in the text and modeling analysis. It is not clear how the effect of clumping is incorporated in the ODE model. As it stands, the modeling analysis gives the impression of data fitting and is somewhat disconnected from the central message (see point 1).

Response: Thank you for pointing this out. We did not include clumping explicitly in the model. However, from all parameters of the model, the rate of drug diffusion into/out of cells (parameter f) is most likely to depend on clumping. Indeed, clumping should reduce the average effective cellular surface area exposed to the extracellular environment, thereby lowering drug influx. Therefore, upon fitting the same model to both clumpy and unicellular data, we tried inferring the effect of clumping from changes in the parameter f versus other parameters. The fits indicated that (i) the drug in/outflux rate constant f was generally lower in the clumpy TBR1 versus the unicellular TBR1 Δ a strain, whereas the inferred f values were similar in the unicellular pair of

BY4742 versus BY4742 Δ a lab strains; (ii) the threshold q that intracellular drug concentration needs to reach for growth inhibition (i.e., tolerance to the drugs) tended to be higher for Δ a strains; (iii) the detox rate p tended to be higher in the AMN1-containing versus AMN1-deleted strains. Other parameters changed in ways that were less consistent. Overall, this suggests that AMN1 affects resistance in a variety of ways, and clumping is only one of the multiple resistance mechanisms that further studies will need to disentangle. We added the above information into the revised Results section.

4. Aspects of the ODE model formulation are also somewhat unclear. For example, the authors assume growth of all their strains exhibit an Allee effect. What's the major basis for this assumption and how critical is this assumption for the central points? It appears that one rationale is the including the effect allows better fit – if this is the major rationale, it's good to make it clearer. Does the magnitude of the Allee term depend on the clumping capability of the strains tested?

Response: Thank you for raising this point. There is recently published evidence for an Allee effect in budding yeast, see Saurabh R. Gandhi, Kirill S. Korolev, and Jeff Gore, PNAS 116 (47) 23582-23587 (2019). Also, as the Reviewer surmised, including the Allee effect allowed better fits, especially for growth curves without drugs. The Allee effect indeed tends to be more apparent in the clumping strain, with a potential contribution from gravitational settling in the plate reader, but is noticeable even in more intensely shaken cultures. Nonetheless, the Allee effect is not the focus of this manuscript, so we did not engage in its detailed investigation. We have now clarified this in the main text and updated the Supporting Information accordingly.

Also, there appears to be some discrepancy in how it's modeled: in Equation 1, the density-dependent growth rate is modeled as $(N+C)$, whereas in Section 2.3 (Supp), it's modeled as $(N-C)$. Is this intended or the result of a typo?

Response: Thank you for catching this! It is indeed unintended (it is a typo). The fits can be done both ways, and they result in Allee parameters of opposite signs, as expected. We have corrected the equations to make them consistent.

5. In equation 6, why is drug uptake not considered? I would expect a term proportional to the first term in Equation 4, but with an opposite sign.

Response: This is an excellent point, which we appreciate very much. We omitted drug uptake from Equation 6 because we considered it negligible compared to the total amount of drug and its degradation within the whole extracellular growth medium, which we expected to have a much larger volume than the cell interiors. This approximation does indeed apply initially, when cell numbers are low, but probably not later, at high cell densities. We have redone the fits with the drug uptake included in Equation 6 (see Figure 6). These details are now reflected in the Supporting Information, Section 2.4.

6. In Fig. 6E, it appears that 'f' (the drug influx rate) either does not change or gets smaller with increasing H₂O₂ or AmB and becomes larger with the increases in the other two drugs in the AMN1 deletion mutant. What should we have expected a priori?

Response: Thank you for the comment. These trends are no longer apparent in the parameter fits based on the updated model (Figure 6).

Was there prior evidence that clumping affects penetration of these different stressors differently?

Response: There was no prior evidence on penetration of these drugs into yeast clumps. Yeast cells in the interior of multicellular flocs were reported to have slower stressful chemical influx by Smukalla et. al, *Cell Cell* 135(4):726-37 (2008). Similar observations were made for circulating tumor cell clusters or cancer cell spheroids, see Hamilton et al., *Cancer Drug Resistance* 2:762-72 (2019). Therefore, we assumed that clumping might similarly reduce drug influx, partly by reducing the effective cell surface area exposed to the extracellular environment.

7. 'Growth curve reshaping' is a vague term and repeatedly used without a clear introduction/definition of it.

Response: Thank you very much for the suggestion. We defined growth curve reshaping as a significant drop in the AUC versus stress-free conditions, due to the drugs "altering the number or slope of growth phases", in the "Loss of AMN1 impairs TBR1 growth in stressful conditions" section of the Results.

8. Line 113: It would be helpful to provide at least a brief definition of the 'normal settings' in the main text.

Response: Thank you for this comment. We have replaced "normal settings" with "without stress" in the revised manuscript. We use this term now to distinguish stress-free conditions from other growth conditions where stressors were added.

9. Authors mention a base level growth even with 0% glucose in the experiments and attribute that to intracellularly stored sugar. That makes me wonder what the authors found for the Allee threshold (C) for 0% glucose.

Response: We have addressed this comment by growing the cells in YPD and then resuspending them in 0% glucose (YP) medium. We still observed a weak Allee effect by growth curve fitting, see the revised Supporting Information, section 2.2. The cells managed to maintain growth for at least 24 hours after being transferred into YP.

10. For the equations along Lines 795 and 796, why doesn't the toxicity mediated killing of cells contribute to nutrients?

Response: This is an interesting suggestion. We considered this suggestion, but ultimately abandoned it because we are uncertain about the detailed outcomes of cell killing. Specifically, if the cells completely lyse and disintegrate then their contents might reenter the medium, but not otherwise. Moreover, the contents potentially reentering the growth medium could be complex and more difficult to utilize than simple sugars. Due to all these complexities, we did not implement this into the model.

11. What would have they seen if they had tested higher concentrations of FLC? My sense is that the tested concentrations might be too low, although they are somewhat recommended by literature (as claimed by the authors).

Response: Thank you for your question. We ran “kill curve” experiments to address this comment. We increased the FLC concentration over 3-fold above the current highest concentration (150 $\mu\text{g}/\text{mL}$) and observed no growth above about 250 $\mu\text{g}/\text{mL}$ (please see the figure below, where the concentrations of FLC added to YPD growth medium are shown in $\mu\text{g}/\text{mL}$). We ultimately selected the upper limit of this stressor as the dose that generated a some visible effect on the two TBR1 strains (150 $\mu\text{g}/\text{mL}$), which became even more pronounced upon resuspension.

12. FLC starts showing its “growth reshaping” effects only once after cells are resuspended in the same type of media before stationary phase. That resuspension also means reducing the cell density. So, might protection from FLC have a cell density-dependent component too in addition to an initial delay? Would it be possible to lower the initial cell density instead of resuspension in order to give cells more time under the FLC treatments tested?

Response: Thank you for this comment. We have tested the effect of the same FLC concentrations starting from lower cell densities (0.005, 0.01, 0.025 OD) than the initial density (0.05 OD) (see the figure below). At lower cell densities, the reshaping resembled what we

observed upon resuspension with a more pronounced effect on the unicellular AMN1 mutant strain (still visible at 0.01 OD), which indeed supports your hypothesis.

A related question, I see in Fig. S11 that resuspension was done after 10 hours and into fresh media containing the original concentration of the drug. Do the authors have any idea about how much drug degradation is going on during those 10 hours?

Response: Thank you for this question. Although FLC drug degradation is media- and temperature dependent, the literature suggests a half-life of 30 hours in plasma (DrugBank Accession Number: DB00196). Among other evidence, the drug penetration time in *Candida* biofilms is ~6 hours. with no change in cell viability between 6-24 hours (indicating modest degradation within a day and explaining the delayed drug effect), see Al-Fattani & Douglas, *Antimicrob. Agents Chemother.* 48(9): 3291–3297 (2004). FLC has not been tested on *S. cerevisiae* TBR1 strain prior to this study. We estimated a degradation rate of ~0.225/hr for FLC, implying that >10% of the drug should still be present after 10 hours. Additionally, we would also like to point out that the elimination of intracellular toxicity is decoupled from drug degradation, so the cells could still be suppressed much after the drug is degraded.

15. Figure 1A-C: it would be nice to have scale bars.

Response: Thank you for pointing this out. The scale bars are now added.

16. Figure 1 F and G: I could not figure out which data are for glucose and which data are for galactose. Probably, galactose data is only in Supp, and this caption needs to be corrected.

Response: Thank you for catching this. Indeed, the galactose data is only shown in Supporting Information, and this caption was rewritten accordingly.

17. Figure 2A: legends for the strains are missing

Response: Thank you for catching this. The legends were added.

18. Fig. 2F: Breakpoint labels are not displayed in the figure.

Response: Thank you, the labels were added.

Also, why was not a death phase considered in labeling the linear pieces of the log (OD600) curves, although it clearly appears in some of the conditions in Fig. 2A-D?

Response: Thank you. For consistency, we decided to call the phase prior to regrowth “adaptation phase”, even if cells were apparently dying.

19. Fig. S15-18: In some many cases (but not all) the results show that a tiny and temporary increase in D (intracellular drug concentration) succeeded by a multiple fold higher and much longer lasting increase in toxicity. I wonder how justifiable this is.

Response: Thank you for the comment. We would like to point out that the accumulation of intracellular toxicity acts as an amplification factor, and elimination of intracellular toxicity is decoupled from drug degradation. Therefore, if the rate of toxicity accumulation is large and toxicity elimination is low, then the cells could still experience high toxicity and be suppressed much after a small amount of drug appears and degrades. While the exact mechanisms of toxicity are unknown and complex, the level and duration of effects such as damage to proteins, RNA and DNA could largely exceed and last well beyond drug elimination.

20. Figs. S19: Please specify which data belong to which stressor. I assumed it is in the order of H2O2, AmB, CASP, and FLC from left to right.

Response: Thank you for noticing this. The stress labels are now added accordingly.

Reviewer #2 (Remarks to the Author):

This is a very thorough and insightful investigation of how a clumping gene (AMN1) impacts adaptation to and growth in the presence of different stresses. A few suggestions would improve the quality of this manuscript:

1. The authors should discuss the fact that the 2 ergosterol-targeting antifungals have different effects on fungal cells (one is static, other is cidal)

Response: Thank you for the valuable recommendation. This was addressed for both AmB and FLC in the following statements in Results and Discussion:

“While considered fungicidal, AmB could also exhibit static activity at sub-MIC₉₀ doses⁶¹.”

“Accounting for the previously noted resistance threshold value (64 µg/mL)⁶⁵ and the static and cidal effects observed below and above this value⁶⁶, we characterized growth curve reshaping by inoculating TBR1 and TBR1Δa in 0, 50, 75, 100, 125, and 150 µg/mL solutions of FLC in YPD.”

“Many antifungals used in clinic have dose- and strain-dependent static and cidal effects (e.g., ergosterol-targeting drugs such as AmB and FLC)^{60, 78}, which complicates the cellular resistance interpretation. Detailed, quantitative investigation of drug effects and their mechanisms would standardize their usage in clinic and laboratory research.”

2. The authors should verify that the 2 TBR1 strains investigated don't show different clumping/clustering patterns upon exposure of the different stress agents.

Response: Thank you for the recommendation. As shown in Figure S6, TBR1 and TBR1Δa showed little change in clump size upon all tested drug conditions. No unexpected single- or multi-cell morphology changes were observed. The microscopy data for each strain and condition imaged after the growth experiment will be made available online.

3. A quantification of clumping of the the 2 BY4742 strains (similar to that shown in Fig1A-E) should be included to strengthen the author's conclusions regarding the roles of AMN1 in different strain backgrounds

Response: Thank you for the great suggestion. We included clump size data for the BY4742 strains in Figure 1D (between the other unicellular strains) and added microscope images of these strains to the Supporting Information (Figure S22). We did not include cell size histograms for the BY4742 strains in Figure 2E since they were closely overlapping with TBR1Δa. Nonetheless, we will make the data available online.

REVIEWERS' COMMENTS:

Reviewer #1 (Remarks to the Author):

The authors have fully addressed my raised issues and I support the publication of the paper in the journal.